# IN-CONTEXT LEARNING AT REPRESENTATION LEVEL VIA UNLABELED TEXTS

## ABSTRACT

Large language models (LLMs) have exhibited impressive capability of In-Context Learning (ICL), where LLMs perform relatively complicated tasks beyond the pre-training objective by conditioning on the given demonstrations. Nevertheless, ICL introduces two gaps between pre-training and inference: *label appearance* (presence of inserted labels in the demonstrations) and *weak semantic relevance* (independently sampled demonstrations exhibit less semantic coherence compared to consecutive text segments in pretraining corpora). We propose a new inference method that only use unlabeled inputs from the test set and label space. In this method, we extract the representations of the demonstrations inputs independently and fuse them to reshape the representation of the test input for inference. Interestingly, without access to labels, our method outperforms traditional ICL with extra information of gold labels. Furthermore, our method allows small models to outperform the zero-shot performance of models that are twice their size (e.g., GPT-Neo-2.7B surpasses Llama2-7B, and Llama2-7B outperforms Llama2-13B). Our code will be available at this [1].

## 1 INTRODUCTION

One of the representative characteristics of generative large language models (LLMs), e.g., GPT-3 (Brown et al., 2020), Llama2 (Touvron et al., 2023), and Gemini (Team et al., 2023) is their in-context learning (ICL) capabilities. Through task-specific input-output examples, large language models can "learn" to accomplish various tasks beyond the pre-training objective (Dong et al., 2022). Despite the improved down-stream performance compared to zero-shot inference, ICL introduces gaps between pre-training and inference in different aspects. The goal of LLM pre-training is to maximize the likelihood of each **next token** given its **preceding context tokens**, while ICL inference forces LLMs to predict **the output of downstream tasks** conditional on the **given demonstrations** that are not involved in pre-training.

Existing works have recognized the above **target discrimination** between pre-training and ICL and presented a few strategies accordingly. Chen et al. (2022) first propose meta-learning to learn in-context examples with task instructions. Min et al. (2022a) expand the scope of the experiment by covering more diverse tasks without task instructions. However, there are still two gaps between pre-training and ICL that have not been fully discussed. **Label appearance:** Compared to the texts that are not related to a specific task during pre-training, the input-label mapping in ICL inserts additional task information. **Weak semantic relevance:** Unlike the coherent texts used in pre-training, the ICL demonstration examples are not necessarily semantically relevant. Nevertheless, previous ICL research focuses on **what and how** input-label mapping information is utilized (Kossen et al., 2023; Pan et al., 2023), but neglects **when** the positive effect brought by ICL exceeds the negative influence of pretraining-inference gaps.

Given this, we first explore the two gaps caused by demonstration examples, i.e., the aforementioned *label appearance*, and the *weak semantic relevance*. We then calibrate *when bridging these two gaps surpasses the improvement from ICL*. To eliminate the negative effects of different gaps, we propose a new ICL paradigm: conducting in-context learning at the representation level via unlabeled texts.

---

[1] https://anonymous.github.com

Table 1: All the datasets used in the experiments.

| Dataset | Source | Task | Test Size |
|---------|--------|------|-----------|
| **RTE** | News, Wikipedia | Natural Language Inference | 277 |
| **MRPC** | Miscellaneous | Paraphrase Detection | 1,725 |
| **COLA** | Miscellaneous | Grammar Error Detection | 1,043 |
| **MNLI** | Miscellaneous | Natural Language Inference | 9,815 |
| **SST2** | Movie Reviews | Sentiment Analysis | 872 |
| **ACL** | Academic Papers | Citation Intent Analysis | 139 |
| **MUSIC** | Music Description | Music Genre Identification | 1,010 |
| **PHRASE** | Financial News | Sentiment Analysis | 2,264 |

**Contributions** Throughout this paper, we mainly (1) propose a new ICL paradigm that conducts in-context learning at the representation level via unlabeled texts from the test set. (2) demonstrate that our method outperforms zero-shot significantly over eight datasets from multiple sources and surpasses traditional in-context learning. (3) disclose the limitations along with the potential reasons and solutions for further performance improvement.

**Important Observations** Despite being at its preliminary stages, this work offers several vital observations regarding in-context learning, which can serve as valuable references for future research. (1) When working with specific-domain datasets, the positive impact of task-related labels outweighs the negative effects of the label appearance gap. However, the opposite is true for general-domain datasets. (2) The input-label mapping information provided by demonstrations is considerably more beneficial for specific-domain datasets than general-domain datasets. (3) Conditioning in-context learning on the independent representations of demonstration inputs proves more effective in bridging weak semantic relevance than conditioning it on the concatenation.

In the rest of this paper, we first launch a preliminary study of the two gaps: *label appearance* and *weak semantic relevance* in Section 2. Building on the analysis, we propose a new in-context learning paradigm in Section 3 and extensive experiments for our method in Section 4. Section 5 provides the background for in-context learning and the efforts put into understanding it.

## 2 PRELIMINARY ANALYSIS

In this section, we will first show that the absence of labels has little harm to the performance, which implies that "unlabeled ICL" works reasonably well. Next, we analyze a potential weakness of unlabeled ICL, and propose ideas to improve the performance with unlabeled demonstration inputs.

### 2.1 ANALYSIS SETTINGS

**Datasets** We conduct extensive experiments over 8 datasets, including five popular datasets of previous ICL research (Ye et al., 2023; Cheng et al., 2023; Li et al., 2023b), i.e., MRPC (Dolan et al., 2004), COLA (Warstadt et al., 2019), MNLI (Williams et al., 2018), RTE (Wang et al., 2018), SST2 (Socher et al., 2013), and three new datasets, i.e., ACL (Bird et al.), PHRASE (Malo et al., 2014), MUSIC (Wu et al., 2023), to cover more scenarios. Concretely, RTE is collected from Wikipedia that consist of universal world knowledge; MRPC, MNLI, and COLA involve miscellaneous data sources, while the rest of the datasets are constructed from a specific domain, e.g., movie review, academic paper, music, and finance. Hereafter, we refer datasets from general world knowledge like Wikipedia and miscellaneous sources to the general domain category and the left specific data source as the specific domain category. More details about the datasets and tasks are in Table 1.

**Backbones** The analysis experiments involve five widely acknowledged language models of different sizes, including GPT-Neo-2.7B (Black et al., 2021), Mistral-7B (Jiang et al., 2023) Llama2-7B, Llama2-13B (Touvron et al., 2023), and GPT3.5-Turbo-Instruct[2] by OpenAI's public API.

**Evaluation** According to common practices (Brown et al., 2020; Rubin et al., 2022; Ye et al., 2023), we turn the discrete label into a description such as "The review is positive" in SST2, add it to the beginning of the test input (e.g., "Hate it." in the below) as different inputs to language models, compare the LM likelihood of each choice, and choose the one with the maximum likelihood.

---

[2]https://platform.openai.com/docs/guides/text-generation

Table 2: Results of Topk-ICL and Topk-ICL without labels. $\Delta$ means the improvement of w/o labels.

| Model | Methods | General-Domain | | | | Specific-Domain | | | |
|-------|---------|------|------|------|------|------|------|--------|-------|
| | | MRPC | COLA | MNLI | RTE | SST2 | ACL | PHRASE | MUSIC |
| *GPT-Neo-2.7B* | Topk-ICL | 33.91 | 67.11 | 39.03 | 46.57 | 85.44 | 24.46 | 87.28 | 28.51 |
| | w/o labels | 66.49 | 69.13 | 36.51 | 50.18 | 71.67 | 5.04 | 32.55 | 31.78 |
| | $\Delta$ | $+32.58$ | $+2.02$ | $-2.52$ | $+6.14$ | $-13.77$ | $-19.42$ | $-54.73$ | $+3.27$ |
| | **Average** | **$+9.56$** | | | | **$-21.16$** | | | |
| *Llama2-7B* | Topk-ICL | 36.35 | 32.89 | 37.12 | 51.26 | 81.42 | 25.9 | 80.79 | 33.56 |
| | w/o labels | 66.49 | 35.76 | 34.72 | 50.90 | 62.39 | 12.23 | 68.02 | 36.93 |
| | $\Delta$ | $+30.14$ | $+2.87$ | $-2.40$ | $-0.36$ | $-19.42$ | $-13.67$ | $-12.77$ | $+3.37$ |
| | **Average** | **$+7.56$** | | | | **$-10.62$** | | | |

$$[\texttt{The review is positive. Hate it.}] \implies 0.3 \times$$
$$[\texttt{The review is negative. Hate it.}] \implies 0.9 \checkmark$$

We investigate the most commonly used in-context learning setting: Topk-ICL, where the test input and candidates are encoded by a pre-trained encoder, and those with the highest cosine similarity to the test input are selected as demonstrations. The number of demonstrations is 16, which is well-studied in ICL, considering LLMs' constrained context window size. The encoder here is all-mpnet-base-v2 (Reimers & Gurevych, 2019), which is widespread and available in Huggingface Transformers (Wolf et al., 2020)[3]. To ensure reproducibility, we set the random seed to 42. The evaluation metric used is accuracy.

## 2.2 THE EFFECT OF LABEL-APPEARANCE

***When processing specific-domain datasets, the positive effect of task-related labels exceeds the negative influence of the label-appearance. However, for general-domain datasets, it is just the opposite.*** We conduct a controlled experiment with/without labels in top-ICL to analyze the possible effect of label-appearance. Table 2 reveals that eight datasets exhibit varying performance trends when labels are removed from demonstrations. For instance, some datasets like MRPC benefit from the change while others like PHRASE suffer greatly. However, when viewing datasets in groups, the trend is clear: removing labels boosts the performance of general-domain datasets while conversely reducing the performance of specific-domain datasets. This discovery shows that labels are far more critical in specific-domain datasets than in general-domain.

***The input-label mapping information provided by demonstrations benefits specific-domain datasets much more than general-domain.*** The above analysis reveals that labels play a vital role in specific-domain datasets while are less critical in general-domain datasets. We suppose that ICL benefits from the input-label mapping information when processing specific-domain datasets. Following the previous work (Min et al., 2022b; Pan et al., 2023), we study the performance difference when shuffling labels in demonstrations to analyze the effect of the correct input-label mapping information. According to Figure 1, when modeling random labels, the performance of four language models in all the datasets decreases. The reduction in performance indicates that the correct input-label mapping information benefits all the models and datasets. However, the performance drops much more in specific-domain datasets than general-domain datasets, implying that correct input-label mapping is much more needed in specific-domain datasets.

## 2.3 WEAK SEMANTIC RELEVANCE AND HOW TO BETTER UTILIZE UNLABELED INPUTS

Unlike accepting ***single coherent text*** in language models pretraining, in-context learning takes in the ***concatenation of multiple demonstrations*** which are not necessarily relevant. This observation leads to a new design of ICL, as we illustrate below.

Let us revisit the previous finding in a more mathematical way. For the standard ICL with gold input-output pairs, the inference process of a LLM can be expressed as follows:

$$\hat{y}_{\text{test}} = \text{LLM}(x_1, y_1; x_2, y_2; \ldots; x_m, y_m; x_{\text{test}}). \tag{1}$$

---

[3]https://huggingface.co/sentence-transformers/all-mpnet-base-v2

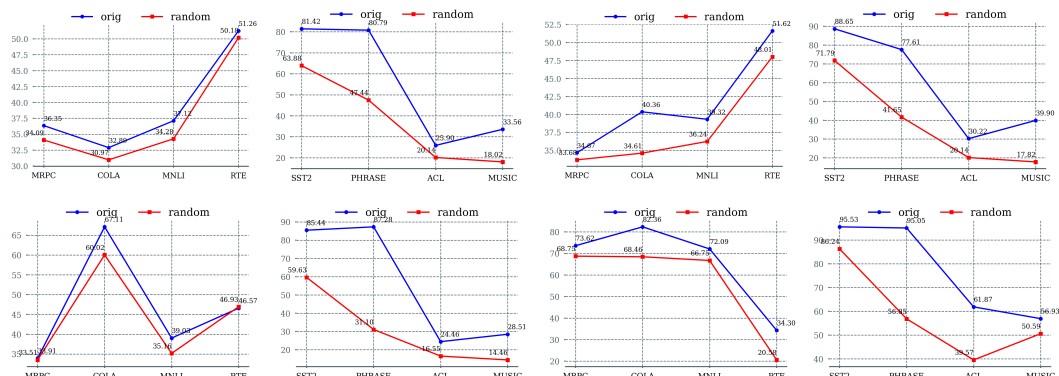

Figure 1: We report the reduction in the performance brought by random labels (general-domain vs specific-domain). From left to right, they are Llama2-7B (2.02 vs 18.05), Llama2-13B (3.36 vs 21.24), GPT-Neo-2.7B (2.93 vs 25.99), and gpt-3.5-turbo (9.46 vs 19.03).

In the absence of labels for the demonstrations, the prompt comprises demonstration inputs and a test sample, which allows us to describe the process as follows:

$$\hat{y}_{\text{test}} = \text{LLM}(z_1, z_2, \ldots, z_m; x_{\text{test}}), \tag{2}$$

where $z_1, \ldots, z_m$ represent the unlabeled inputs. "unlabeled ICL" works well in Section 2.2.

**The gap between pretraining and inference.** We observed a poential weakness of the "unlabled ICL" in equation 2. LLMs are pre-trained on "coherent text" $(u_1, u_2, \ldots, u_s; u_{s+1})$ extracted from an article or a file, where the inputs $u_1, u_2, \ldots, u_s, u_{s+1}$ have strong semantic dependence. For instance, the sentence "*Apples are juicy and delicious, and many kids like to eat them*" may appear in the pre-trained corpus, and the words exhibit strong semantic dependence. In contrast, in unlabled ICL equation 2, the inputs $z_1, z_2, \ldots, z_m, x_{\text{test}}$ are (independently) sampled from a certain distribution, instead of from an article, thus their semantic dependence is weak. For instance, when $m = 1$, $z$ and $x_{\text{test}}$ can be $z =$ "*apple*", $x_{\text{test}} =$ "*car*", which exhibits weaker semantic dependence.

To design a better way to utilize the demonstration inputs, we briefly analyze the mechanism of unlabeled ICL. In the equation equation 2, we suspect that the demonstration inputs $(z_1, \ldots, z_m)$ serve as the contextual information that helps LLM better "understand" the query input $x_{\text{test}}$. Nevertheless, processing the concatenation $(z_1, \ldots, z_m; x_{\text{test}})$ by LLM may not be the best way to utilize the context information of $z_1, \ldots, z_m$ since LLMs are not trained to handle $m$ consecutive samples drawn from an independent distribution. One possible path for design is to consider various ways of combining demonstration inputs in the prompt (i.e., prompt design). In this paper, we aim to explore the representation space, and develop better methods to manipulate the representations of $z_1, \ldots, z_m, x_{\text{test}}$, hoping that this may provide some improvement.

**New Idea: Processing Representations of Demonstration Inputs and Test Input Independently** How to improve unlabled ICL? Our idea is the folloiwng: Since the demo inputs $z_1, \ldots, z_m$ and $x_{\text{test}}$ are not from a coherent text, they do not necessarily need to be processed as a whole by the LLM. Instead, they can be processed indepedently by the LLM and then combined for inference.

To illustrate the idea and analyze its validity, we use an example of $m = 1$. When $m = 1$, we consider two samples $z$ and $x_{\text{test}}$, which are independently drawn from a certain distribution. The unlabeled ICL can be expressed as $y_{\text{test}} = \text{LLM}(z; x_{\text{test}})$. For notation simplicity, we denote $A = x_{\text{test}}$, and $B = z$. It is not easy to analyze the effect of an LLM, and we simply analyze one layer of self-attention. This leads to `Method 1`: computing the self-attention output of the concatenated sequence $(B; A) = (z; x_{\text{test}})$. As an alternative, we anlyze another method which takes the representation of $A$ and $B$ separately. `Method 2`: first computing the self-attention output of A and B respectively, then applying cross-attention between A and B. We compare the final representation of all the tokens in $A$.

The final representation of the $i$-th token in Method 1:

$$a'_i = \sum_{j=1}^{N_A} \alpha_{ij} a_i + \sum_{m=1}^{N_B} \beta_{i, N_A + m} b_m, \tag{3}$$

where $a_i$ and $b_m$ represent the $i$-th and $m$-th token in A and B, $N_A$ and $N_B$ represent the total length of A and B, and $\alpha$ and $\beta$ denote the attention score of the $i$-th token when attending $A$ and $B$.

In the second setting, the final representation of the $i$-th token is computed as follows:

$$\tilde{a}_i = \sum_{j=1}^{N_A} \alpha_{ij} a_i, \quad \tilde{b}_j = \sum_{m=1}^{N_B} \delta_{jm} b_m, \quad a_i'' = \sum_{m=1}^{N_B} \gamma_{im} \tilde{b}_m, \tag{4}$$

where $\delta, \gamma$ denote the attention score in computing self-attention in B and cross-attention between A and B. In the first setting, the weakly-relevant information from B is directly included in the attention context for every token in A, which can add weak semantic relevance to the representation of A's tokens. A's self-attention is computed independently of B, preserving A's original context. The cross-attention allows the $i$-th token to selectively incorporate information from B given the context of A, potentially reducing the impact of weakly-relevant information from B. Thus, the second setting mitigates weak semantic relevance better than the first.

With the above analysis, we propose representing the demonstration input and test input separately, rather than concatenating them. Next, we briefly discuss how to combine the independent representation of the demonstration input $z$ and the test sample $x_{\text{test}}$. The further details about how to handle multiple demonstration inputs are provided in Section 3.

**How to Utilize The Representation of Demonstration Input?** There is a remaining question: how to combine the independent representation of the demonstration input $z$ and the test sample $x_{\text{test}}$? We would like to utilize the context information of $z$ to better represent $x_{\text{test}}$.

We borrow insight from the attention mechanism: we treat $x_{\text{test}}$ as a "query" and $z$ as "keys" and "values", and then utilize the relevant information from the "keys" to reconstruct a representation of $x_{\text{test}}$. The reconstructed representation incorporates the contextual information of $z$.

Given query, key, and value matrices $\mathbf{Q} \in \mathbb{R}^{n \times d}$, $\mathbf{K} \in \mathbb{R}^{m \times d}$, and $\mathbf{V} \in \mathbb{R}^{m \times d}$, the output of an attention layer can be defined as follows, where $\tau$ is the temperature, $s(\cdot, \cdot) \in \mathbb{R}$ is a scalar function, $\boldsymbol{q}_i, \boldsymbol{k}_i, \boldsymbol{v}_i \in \mathbb{R}^d$ denotes the $i$-th row of $\mathbf{Q}$, $\mathbf{K}$ and $\mathbf{V}$, and $n$, $m$ are the number of rows in $\mathbf{Q}$ and $\mathbf{K}$.

$$f_A(\mathbf{Q}, \mathbf{K}, \mathbf{V}) = \begin{bmatrix} \left(\sum_{j=1}^m \alpha_{1j} \boldsymbol{v}_j\right)^\top \\ \left(\sum_{j=1}^m \alpha_{2j} \boldsymbol{v}_j\right)^\top \\ \vdots \\ \left(\sum_{j=1}^m \alpha_{nj} \boldsymbol{v}_j\right)^\top \end{bmatrix} \in \mathbb{R}^{n \times d}, \text{ where } \alpha_{ij} = \frac{\exp(s(\boldsymbol{q}_i, \boldsymbol{k}_j)/\tau)}{\sum_{l=1}^m \exp(s(\boldsymbol{q}_i, \boldsymbol{k}_l)/\tau)}. \tag{5}$$

For each query vector $\boldsymbol{q}_i$, the output $\hat{q}_i$ is a weighted sum of the rows of $\mathbf{V}$ (i.e., a linear combination of the rows of $\mathbf{V}$). This implies that the query $\boldsymbol{q}_i$ is mapped onto the vector space spanned by the value vectors $\boldsymbol{v}_1, \boldsymbol{v}_2, \cdots, \boldsymbol{v}_m$. Intuitively, this means that the reconstructed $\hat{q}_i$ is a new vector that incorporates the context of the value vectors in the representation of the query.

This observation inspires the idea that we can build a new representation of the test input in the following way. For each token in the test input, we map the representation of it onto the space spanned by the representation of the demonstration input to obtain a new representation vector. In essence, this method performs in-context learning at the representation level.

## 3 METHOD

In this section, we formally introduce the proposed new ICL paradigm that conducts in-context learning at the representation level via unlabeled texts. The overview of our method is presented in Figure 2. Suppose there are $T$ test samples in the test dataset $\mathcal{D}$ for a certain task. The goal is to provide a prediction for each sample in the test dataset.

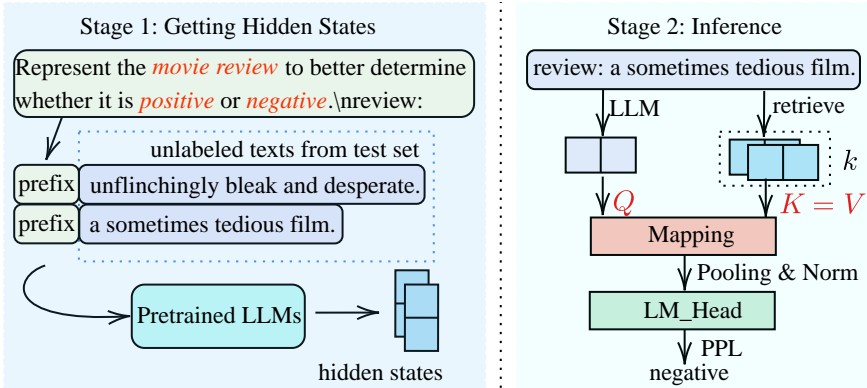

Figure 2: Overview of our method.

**Utilizing Other Unlabeled Samples** In classical zero-shot inference, the prediction for each test sample is independent of other test samples, and the prediction can be formulated as

$$y_{test,i} = f_\theta(x_{test,i}), i = 1, 2, \ldots, T, \tag{6}$$

where $f_\theta$ indicates the learned neural network. In our method, the prediction for each test sample is based on other $k$ relevant test samples excluding itself, and the prediction can be formulated as

$$y_{test,i} = g(x_{test,i}; \underbrace{x_{test,p}, \ldots, x_{test,q}}_{k}), i = 1, 2, \ldots, T, \tag{7}$$

where $g$ is a certain process that we will describe next.

**Step 1: Obtaining Feature Vector for Each Test Sample** For any given task, we establish a task description $\mathcal{T}$ that includes the basic input units and labels related to the dataset. For instance, the description for SST2 Socher et al. (2013) is "Represent the *movie review* to better determine whether it is *positive* or *negative*." We concatenate the description with the test input $x = (x_1, x_2, ..., x_n)$ where $x_i$ denotes the $i$-th token of $x$, and feed the concatenation into LLMs to obtain hidden states.

$$\mathbf{H}_1, \mathbf{H}_2, ..., \mathbf{H}_n = \text{LLM}([\mathcal{T}; x]) \tag{8}$$

$$\mathbf{H}_t = [\boldsymbol{h}_t^{(1)}; \boldsymbol{h}_t^{(2)}; \cdots; \boldsymbol{h}_t^{(L)}], \tag{9}$$

where $\boldsymbol{h}_t^{(l)} \in \mathbb{R}^d$ represents the hidden state of the $t$-th token at the $l$-th layer, and $L$ represents the number of layers in the LLM. We employ three well-known pooling strategies to attain the feature vector from the set of hidden states, for each token. `Last`: pool the hidden state of the last layer; `Last-Two`: pool the hidden states of the last two layers and average across the last dimension; `First-Last`: pool the hidden states of the first and last layer and average across the last dimension. Next, we will use `Last` pooling strategy as an example to explain our method, in which case the feature vector for the $t$-th token can be denoted as $\boldsymbol{h}_t = \boldsymbol{h}_t^{(L)}$.

**Step 2: Reconstructing the Feature Vector** For any test input $x$ in the test dataset, we will reconstruct its test feature vector as follows. First, we identify the $k$-th most relevant test inputs $z_1, \ldots, z_k$ based on a certain retrieval algorithm (for instance, BM25). For each test sample $z_s = (z_{s1}, \ldots, z_{sn})$ which consists of $n$ tokens, $1 \le s \le k$, we denote the corresponding feature vector obtained in the previous stage as $\boldsymbol{h}_{s1}, \boldsymbol{h}_{s2}, \cdots, \boldsymbol{h}_{sn}$, where $\boldsymbol{h}_{sj}$ corresponds to $z_{sj}$.

Second, we reconstruct the feature vectors of $x$ utilizing the feature vectors of the retrieved test inputs $z_1, \ldots, z_k$. Suppose the corresponding feature vectors of $x = (x_1, \ldots, x_n)$ are $\boldsymbol{h}_{01}, \boldsymbol{h}_{02}, \cdots, \boldsymbol{h}_{0n}$, we compute new feature vectors $\boldsymbol{h}_{s;i}$ as follows:

$$\boldsymbol{h}_{s;i} = \sum_{j=1}^{n} \alpha_{ij} \boldsymbol{h}_{sj}, \text{ where } \alpha_{ij} = \frac{\exp(s(\boldsymbol{h}_{0i}, \boldsymbol{h}_{sj})/\tau)}{\sum_{l=1}^{n} \exp(s(\boldsymbol{h}_{0i}, \boldsymbol{h}_{sl})/\tau)}. \tag{10}$$

Here $\boldsymbol{h}_{s;i}$ denotes the attended result of $h_{0i}$ with the context of $h_{s1}, h_{s2}, \cdots, h_{sn}$, and all the score functions $s(\cdot, \cdot)$ are listed in Table 3.

Third, for the $i$-th token, after obtaining $k$ new feature vectors $\boldsymbol{h}_{s;i}, s = 1, \ldots, k$, we take the average of them, and then take a weighted sum of this average and the original test input feature vector.

Table 3: All the mapping methods used in the experiments. $\bar{x}$ represents the standard score format of $x$, while $\| \cdot \|_2$ and $\| \cdot \|_1$ denote L2 and L1 distance.

| Method | Attention | Cosine | Pearson | Euclidean | Manhattan |
|---|---|---|---|---|---|
| $s(\boldsymbol{q_i}, \boldsymbol{k_j})$ | $\frac{\langle \boldsymbol{q_i}, \boldsymbol{k_j} \rangle}{\sqrt{d}}$ | $\frac{\langle \boldsymbol{q_i}, \boldsymbol{k_j} \rangle}{\|\boldsymbol{q_i}\| \cdot \|\boldsymbol{k_j}\|}$ | $\frac{\langle \bar{\boldsymbol{q_i}}, \bar{\boldsymbol{k_j}} \rangle}{d-1}$ | $\|\boldsymbol{q_i} - \boldsymbol{k_j}\|_2$ | $\|\boldsymbol{q_i} - \boldsymbol{k_j}\|_1$ |

$$(\tilde{\boldsymbol{h}}_{0i})_m = 0.4 \cdot \mathrm{mean}\bigg( (\boldsymbol{h}_{1;i})_m, (\boldsymbol{h}_{2;i})_m, \cdots, (\boldsymbol{h}_{k;i})_m \bigg) + 0.6 \cdot (\boldsymbol{h}_{0i})_m, \tag{11}$$

where $(\tilde{\boldsymbol{h}}_{0i})_m$ denotes the $m$-th element in $\tilde{\boldsymbol{h}}_{0i}$. This new feature vector combines the relevant information from the $k$ retrieved test inputs $z_1, \ldots, z_k$, and the target test input $x$.

Finally, we normalize the reconstructed feature vector as follows:

$$\boldsymbol{p}_i = \frac{\tilde{\boldsymbol{h}}_{0i}}{\|\tilde{\boldsymbol{h}}_{0i}\|}. \tag{12}$$

**Step 3: Making Predictions** When making predictions, we add the label description to the test input, same as Section 2.1. After reconstructing the feature vector as $p_i$ for $i$-th token in $x$, we exploit the original lm_head. Then, we compute the LM likelihood for every choice based on the corresponding logits and choose the one with the maximum likelihood.

**Implementation Details** We experiment with five mapping methods described in Table 3. For every mapping method, we explore three main variables of our method: pooling strategies $\in [\text{Last}, \text{Last-Two}, \text{First-Last}]$; the number of the retrieved hidden states $k \in [16, 32, 64]$; the temperature $\tau \in [1, 1.5]$. We initially set $k = 64, \tau = 1$ to identify the optimal pooling strategy. Following that, we adjust the value of $k$, and finally, we tweak the value.

## 4 EXPERIMENTS

### 4.1 COMPARASION SETTINGS

**Baselines** Following the setting in Section 2.1, we experiment with the same datasets and language models. Our method has no access to the training set. Thus, we first compare our method to the zero-shot setting to validate its effectiveness. We also compare our method with traditional in-context learning. According to the convention, we experiment with three learning-free ICL settings: random, bm25, and topk. The number of demonstrations is 16, the same as Section 2.1. `Random-ICL`: the demonstrations are selected randomly without repetition. `BM25-ICL`: we adopt BM25 to obtain scores, and select $k$ demonstrations with the highest scores. We report the best result of five score functions in the following experiments. Our method only utilizes unlabeled texts from the test set while traditional ICL employs input-output pairs from the training set.

### 4.2 COMPARISON WITH ZERO-SHOT

**Broad Improvements in Comparison with Zero-Shot** The results in Table 4 demonstrate that our method outperforms zero-shot consistently, with an average improvement of $11.44\%$ in GPT-Neo-2.7B, $16.49\%$ in Mistral-7B, $17.06\%$ in Llama2-7B, and $12.84\%$ in Llama2-13B. Somewhat surprisingly, all the models benefit from our method, even though they vary in size and belong to different model families. Meanwhile, the results indicate that the gains achieved by our method are consistent across tasks and domains, showcasing its generality.

**Boosting Specific-Domain via Unlabeled Texts** Although our method involves no labels, the improvements in specific-domain are remarkable. The label space information incorporated in representing unlabeled texts accounts for the improvements, demonstrating the importance of label space to in-context learning, consistent with the finding in Min et al. (2022b). The improvements in specific domain suggest that our method can be applied in low-resource scenarios.

**Enhancing Small Models to Beat Bigger Models** hen utilizing our method, the performance of GPT-Neo-2.7B exceeds the zero-shot performance of Llama2-7B even though Llama2-7B is more

Table 4: Our method outperforms zero-shot significantly on average. We show the best improvement over zero-shot and **bold** the best results. The significance level is set to 0.05 according to convention. The p-value is shown in the bracket.

| Method | MRPC | COLA | MNLI | RTE | SST2 | ACL | PHRASE | MUSIC | Average |
|---|---|---|---|---|---|---|---|---|---|
| *GPT-Neo-2.7B (p-value: .047)* | | | | | | | | | |
| **Zero-shot** | **66.67** | 69.13 | 34.06 | 47.29 | 76.95 | 6.47 | 29.24 | 14.55 | 43.05 |
| **Our Method** | 66.49 | **69.13** | 43.75 | 54.87 | 85.44 | 48.92 | 38.60 | 28.71 | 54.49 |
| Δ(Absolute Gain) | −0.18 | +0 | +9.69 | +7.58 | +8.49 | +42.45 | +9.36 | +14.16 | **+11.44** |
| *Mistral-7B (p-value: .022)* | | | | | | | | | |
| **Zero-shot** | 35.19 | 34.04 | 37.17 | **53.07** | 79.82 | 12.95 | 59.63 | 39.60 | 43.93 |
| **Our Method** | 66.49 | 69.13 | 52.28 | 52.71 | 85.32 | 51.08 | 63.47 | 42.87 | 60.42 |
| Δ(Absolute Gain) | +31.3 | +35.09 | +15.11 | −0.36 | +5.50 | +38.13 | +3.84 | +3.27 | **+16.49** |
| *Llama2-7B (p-value: .006)* | | | | | | | | | |
| **Zero-shot** | 57.97 | 30.87 | 34.36 | 48.38 | 63.99 | 12.95 | 46.07 | 32.57 | 40.90 |
| **Our Method** | 66.49 | 69.13 | 40.36 | 53.43 | 87.39 | 43.88 | 62.68 | 40.30 | 57.96 |
| Δ(Absolute Gain) | +9.8 | +38.26 | +6.00 | +5.05 | +23.40 | +30.93 | +16.61 | +7.73 | **+17.06** |
| *Llama2-13B (p-value: .046)* | | | | | | | | | |
| **Zero-shot** | 51.19 | 30.87 | 42.83 | **62.45** | 77.98 | 12.23 | 54.02 | 35.05 | 45.83 |
| **Our Method** | 66.49 | 69.13 | 48.10 | 57.40 | 86.24 | 43.88 | 54.55 | 43.56 | 58.67 |
| Δ(Absolute Gain) | +15.3 | +38.26 | +5.27 | −5.05 | +8.26 | +31.65 | +0.53 | +8.51 | **+12.84** |

than twice as large. This also holds for Llama2-7B and Llama2-13B, suggesting that our method can enable small models to perform even better than larger models that are two times their size, indicating the application potential in real-world problems.

## 4.3 Comparison with Traditional In-Context Learning

**Conducting ICL at Representation and Text Level** We compare the performance of ICL when conditioned on the concatenation of multiple demonstrations (*text level*) to independent representations (*representation level*). As illustrated in Table 5, five untrained mapping strategies all surpass the concatenation way, indicating that our proposed method can mitigate weak semantic relevance better as discussed in Section 2.3, thus leading to improved performance. Interestingly, the cosine and pearson perform almost the same, for they care about the similar relationship. The experimental results suggest that when conditioned on the independent representations of demonstrations, in-context learning more effectively bridges weak semantic relevance compared to when conditioned on the concatenation of multiple demonstrations.

**Surpassing In-Context Learning in General-Domain** We summarize the results of comparing our method with traditional ICL in Table 6. The results demonstrate that our method outperforms traditional in-context learning for all the models in three different settings. Our method with *no labels* beats traditional in-context learning with *input-label pairs* from the training set. The considerable enhancement suggests that conducting in-context learning at the representation level works too, apart from the traditional text level.

**Partly Worse than In-Context Learning in Specific-Domain** We also compare our method with ICL in specific-domain datasets. Table 6 illustrates that our method performs partly worse than traditional ICL in specific-domain datasets especially in PHRASE. This drop is foreseeable since we

Table 5: Results of ICL at text and representation level with unlabeled texts from the test set. We report the improvement brought by different mapping methods.

| Model | Method | General-Domain | | | | Specific-Domain | | | | Avg |
|---|---|---|---|---|---|---|---|---|---|---|
| | | MRPC | COLA | MNLI | RTE | SST2 | ACL | PHRASE | MUSIC | |
| *GPT-Neo-2.7B* | **Text** | 66.49 | 69.03 | 35.67 | 48.38 | 61.35 | 5.04 | 29.81 | 32.87 | — |
| | **Attention** | +0 | +0.1 | +7.2 | +6.49 | +20.19 | +36.69 | +0.49 | −7.62 | **+7.94** |
| | **Cosine** | +0 | +0.1 | +5.3 | +5.41 | +24.09 | +43.88 | +1.29 | −6.14 | **+9.24** |
| | **Pearson** | +0 | +0.1 | +5.31 | +5.41 | +24.09 | +43.88 | +1.29 | −6.14 | **+9.24** |
| | **Euclidean** | +0 | +0.1 | +8.08 | +6.85 | +23.40 | +42.44 | +0.98 | −8.02 | **+9.23** |
| | **Manhattan** | +0 | +0.1 | +7.29 | +6.13 | +23.51 | +42.44 | +8.79 | −4.16 | **+10.51** |
| *Llama2-7B* | **Text** | 66.49 | 35.19 | 34.18 | 49.82 | 70.53 | 6.47 | 64.66 | 40.89 | — |
| | **Attention** | +0 | +33.94 | +5.19 | +3.61 | +15.59 | +30.22 | −2.03 | −1.98 | **+10.57** |
| | **Cosine** | +0 | +33.94 | +6.18 | +3.61 | +16.63 | +34.54 | −1.23 | −1.88 | **+11.47** |
| | **Pearson** | +0 | +33.94 | +6.17 | +3.61 | +16.63 | +34.54 | −1.23 | −1.88 | **+11.47** |
| | **Euclidean** | +0 | +33.94 | +5.22 | +3.61 | +16.28 | +30.22 | −1.98 | −0.59 | **+10.84** |
| | **Manhattan** | +0 | +33.94 | +5.42 | +3.25 | +16.86 | +37.41 | −1.98 | −1.19 | **+11.71** |

Table 6: Our method, utilizing unlabeled texts from **the test set**, outperforms traditional in-context learning that relies on gold input-output pairs from **the training set** on average. We show the improvement over traditional ICL and **bold** the best results. Note that our approach exclusively employs BM25 without incorporating topk retrieval for practical use.

| Method | MRPC | COLA | MNLI | RTE | SST2 | ACL | PHRASE | MUSIC | Average |
|---|---|---|---|---|---|---|---|---|---|
| *GPT-Neo-2.7B* | | | | | | | | | |
| **Our Method** | **66.49** | **69.13** | **43.75** | **54.87** | **85.44** | **48.92** | 38.60 | 28.71 | — |
| $\Delta_{random}$ | +32.87 | +1.15 | +10.15 | +6.86 | +20.53 | +34.53 | −2.04 | +7.42 | **+13.93** |
| $\Delta_{bm25}$ | +32.63 | +2.78 | +6.60 | +4.33 | +8.72 | +15.83 | −34.77 | −0.30 | **+4.48** |
| $\Delta_{topk}$ | +32.58 | +2.02 | +4.72 | +8.3 | +0.00 | +24.46 | −48.68 | +0.20 | **+2.95** |
| *Mistral-7B* | | | | | | | | | |
| **Our Method** | **66.49** | **69.13** | **52.28** | **52.71** | 85.32 | **51.08** | 63.47 | 42.87 | — |
| $\Delta_{random}$ | +32.81 | +35.76 | +17.52 | +5.42 | +4.13 | +35.25 | +7.11 | +1.38 | **+17.42** |
| $\Delta_{bm25}$ | +32.63 | +30.01 | +7.97 | +5.06 | +7.11 | +19.43 | −13.16 | −5.55 | **+10.44** |
| $\Delta_{topk}$ | +32.63 | +30.78 | +7.92 | +5.42 | −1.26 | +19.43 | −22.48 | −1.09 | **+8.92** |
| *Llama2-7B* | | | | | | | | | |
| **Our Method** | **66.49** | **69.13** | **40.36** | **53.43** | **87.39** | **43.88** | 62.68 | 40.30 | — |
| $\Delta_{random}$ | +31.94 | +37.3 | +7.49 | +3.97 | +33.61 | +27.33 | +18.02 | +6.34 | **+20.75** |
| $\Delta_{bm25}$ | +30.08 | +34.33 | +4.99 | +1.44 | +19.27 | +12.94 | −3.75 | +9.8 | **+13.64** |
| $\Delta_{topk}$ | +30.14 | +36.24 | +3.24 | +2.17 | +5.97 | +17.98 | −18.11 | +6.74 | **+10.55** |
| *Llama2-13B* | | | | | | | | | |
| **Our Method** | **66.49** | **69.13** | **48.10** | **57.40** | 86.24 | **43.88** | 54.55 | 43.56 | — |
| $\Delta_{random}$ | +32.69 | +35.48 | +12.94 | +9.39 | +5.05 | +27.33 | +11.84 | +6.13 | **+17.61** |
| $\Delta_{bm25}$ | +31.88 | +27.9 | +10.06 | +6.14 | +4.13 | +11.51 | −8.44 | +6.23 | **+11.18** |
| $\Delta_{topk}$ | +31.82 | +28.77 | +8.78 | +5.78 | −2.41 | +13.66 | −23.06 | +3.66 | **+8.38** |

have found that the input-label mapping information provided by demonstrations benefits specific-domain datasets much more than general-domain in Section 2.2. Additionally, our method relies more heavily on the intrinsic capabilities of LLMs, as it does not incorporate input-label information. For GPT-Neo-2.7B, which is trained on the Pile (Biderman et al., 2022), financial news constitutes a small proportion of the Pile, resulting in the most significant performance decline. Thus, for datasets with which LLMs are unfamiliar, our method is likely to fail due to the absence in input-label information.

## 4.4 ABLATION STUDIES

We conduct ablation studies on our method for better understanding. Although several mapping methods are involved in the experiments, their phenomena are similar. Thus, we discuss the effect of pooling strategies and the number of the retrieved hidden states only with *cross-attention*.

**On the Effect of Pooling Strategies** The choice of pooling strategies plays a role in the quality of the reconstructed representation. Thus, we first compare three popular pooling strategies in Table 7. For GPT-Neo-2.7B, most datasets obtain notable improvement ($\geq 3\%$) by choosing the correct pooling strategy, whereas the pooling strategies have a weak influence on the performance of the remaining three models. Additionally, for GPT-Neo-2.7B, pooling the last layer is the optimal strategy, whereas pooling the first and last layers is the most effective approach for the remaining three models. This suggests that larger LLMs might benefit from the low-level information present in the first layer.

Table 7: Results of choosing different pooling strategies with $k = 64, \tau = 1$.

| Strategy | MRPC | COLA | MNLI | RTE | SST2 | ACL | PHRASEBANK | MUSIC | Average |
|---|---|---|---|---|---|---|---|---|---|
| *GPT-Neo-2.7B* | | | | | | | | | |
| **Last Layer** | 66.49 | 69.13 | 42.35 | 54.51 | 60.89 | 38.85 | 29.77 | 25.15 | **48.39** |
| **First Last Layer** | 66.49 | 69.13 | 38.31 | 54.87 | 81.54 | 17.99 | 30.08 | 24.55 | 47.87 |
| **Last Two Layers** | 66.49 | 69.13 | 35.69 | 50.54 | 73.28 | 7.19 | 24.12 | 17.43 | 42.98 |
| *Mistral-7B* | | | | | | | | | |
| **Last Layer** | 66.49 | 69.13 | 51.14 | 52.71 | 76.83 | 51.08 | 59.32 | 41.68 | 58.55 |
| **First Last Layer** | 66.49 | 69.13 | 51.96 | 52.71 | 82.11 | 51.08 | 62.99 | 42.18 | **59.83** |
| **Last Two Layers** | 66.49 | 69.13 | 51.9 | 52.71 | 81.65 | 51.08 | 62.32 | 42.48 | 59.72 |
| *Llama2-7B* | | | | | | | | | |
| **Last Layer** | 66.49 | 69.13 | 37.79 | 53.43 | 80.96 | 31.65 | 61.57 | 37.62 | 54.83 |
| **First Last Layer** | 66.49 | 69.13 | 39.26 | 53.43 | 86.12 | 36.69 | 62.59 | 38.81 | **56.57** |
| **Last Two Layers** | 66.49 | 69.13 | 38.73 | 53.43 | 85.21 | 32.37 | 62.28 | 38.81 | 55.81 |
| *Llama2-13B* | | | | | | | | | |
| **Last Layer** | 66.49 | 69.13 | 44.58 | 56.32 | 77.29 | 33.81 | 54.51 | 40.0 | 55.31 |
| **First Last Layer** | 66.49 | 69.13 | 45.01 | 56.68 | 81.54 | 31.65 | 51.77 | 42.18 | **55.64** |
| **Last Two Layers** | 66.49 | 69.13 | 43.28 | 55.23 | 86.01 | 43.88 | 45.10 | 38.12 | 54.83 |

**On the Effect of the Number of the Retrieved Hidden States** We next compare the number of the retrieved hidden states. According to the results in Table 8, the number of the retrieved hidden states exhibits minimal influence on performance. We hypothesize that as the number increases, the more irrelevant instances are retrieved for the test set is diverse. This explains why increasing the number of retrieved hidden states does not result in significant improvement.

**On the Effect of Mapping Methods** We compare different mapping methods, which are essential in reconstructing the test hidden state. The results presented in Table 9 indicate that the choice of datasets and models significantly influences the preference for different mapping methods. Even with the same dataset, different models prefer different mapping methods. The underlying reason for this observation is that different mapping methods capture distinct aspects of semantic information.

## 5 RELATED WORK

Large language models (LLMs) such as GPT-3 (Brown et al., 2020) exhibit the ability to do in-context learning (ICL), where the model performs a downstream task simply by conditioning on a prompt made up of input-output examples.

**Understanding How ICL Works** Xie et al. (2021); Jiang (2023); Wang et al. (2023); Zhang et al. (2023); Han et al. (2023) propose that ICL can be formulated as the Bayesian inference. Min et al. (2022b); Wies et al. (2023) observe ICL is more about identifying the task than learning it, recovering the capacity obtained in pretraining. However, Kossen et al. (2023) argue that ICL almost always depends on in-context labels, and can learn novel semantics about tasks. Chan et al. (2022); Hahn & Goyal (2023); Raventos et al. (2023) investigate the factors affecting the emergence of ICL. Razeghi et al. (2022) discover that term frequencies in the pretraining data affect the performance of ICL. Some studies explore the relationship between gradient descent and conducting ICL (Dai et al., 2023; Akyurek et al., 2022; Von Oswald et al., 2023; Shen et al., 2023). Yan et al. (2023) empirically establish a principle that strengthens the relationship between two tokens based on their contextual co-occurrences by investigating the role of surface features in text generation.

**ICL Free of Demonstrations at Instance Level** It is hard to get access to the demonstrations pool for ICL in real-world scenarios. Kim et al. (2022); Chen et al. (2023); Li et al. (2023a) bootstrap LLMs to generate pseudo demonstrations. This approach does alleviate the dependency on demonstrations. However, generation may be uncontrollable and unstable, easily accumulating biases when generating multiple demonstrations. Also, generating pseudo demonstrations is often expensive and slow. In the study by Lyu et al. (2023), the approach involves initially retrieving $k$ unlabeled test instances, assigning random labels to them, and subsequently conducting in-context learning. Both Kossen et al. (2023) and our finding demonstrate that ICL indeed depends on in-context labels, thus assigning random labels can be risky, especially for datasets coming from specific-domain.

## 6 LIMITATIONS & CONCLUSION

**Limitations** To begin with, every step requires prediction in text generation problems, which accumulates latency in adopting our method. Therefore, our work currently only involves text classification problems in the experiment, leaving a gap in text generation. Second, our method is partly worse than traditional in-context learning in specific-domain datasets, showing there is still room for improvement in exploring the relationship between unlabeled texts and the label space information. Last but not least, our method does not incorporate any training; thus the potential of our approach has not been fully explored. We leave all the above for future work.

**Conclusion** We first analyze the effects of label appearance and weak semantic relevance in traditional in-context learning. Building on the analysis, we propose a new ICL paradigm, which conducts in-context learning at the representation level via unlabeled texts. Results over eight datasets coming from general and specific domain and four language models demonstrate that our method exhibits broad and significant improvements compared to zero-shot. Besides, our method with unlabeled texts from the test set surpasses traditional in-context learning with demonstrations from the training set. Furthermore, our method enables small models to perform even better than larger models that are two times their size. Also, our method boosts specific-domain scenarios only with unlabeled texts, showing the potential in real-world problems, which deserves more attention in the future.

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

# A APPENDIX

Table 8: Results of various numbers of the retrieved hidden states with the best pooling strategy observed in Table 7.

| Model | MRPC | COLA | MNLI | RTE | SST2 | ACL | PHRASEBANK | MUSIC | Average |
|---|---|---|---|---|---|---|---|---|---|
| *GPT-Neo-2.7B* | | | | | | | | | |
| $k = 16$ | 66.49 | 69.13 | 42.65 | 54.15 | 81.31 | 41.73 | 30.3 | 25.15 | **51.36** |
| $k = 32$ | 66.49 | 69.13 | 42.39 | 54.87 | 81.42 | 38.85 | 30.08 | 25.25 | 51.06 |
| $k = 64$ | 66.49 | 69.13 | 42.35 | 54.87 | 81.54 | 38.85 | 30.08 | 25.15 | 51.06 |
| *Mistral-7B* | | | | | | | | | |
| $k = 16$ | 66.49 | 69.13 | 52.07 | 52.71 | 82.22 | 51.08 | 62.63 | 41.78 | 59.76 |
| $k = 32$ | 66.49 | 69.13 | 52.03 | 52.71 | 82.22 | 51.08 | 63.21 | 41.98 | 59.86 |
| $k = 64$ | 66.49 | 69.13 | 51.96 | 52.71 | 82.11 | 51.08 | 62.99 | 42.48 | **59.87** |
| *Llama2-7B* | | | | | | | | | |
| $k = 16$ | 66.49 | 69.13 | 39.35 | 53.43 | 85.44 | 35.97 | 62.37 | 38.91 | 56.39 |
| $k = 32$ | 66.49 | 69.13 | 39.29 | 53.43 | 85.78 | 36.69 | 62.54 | 38.71 | 56.51 |
| $k = 64$ | 66.49 | 69.13 | 39.26 | 53.43 | 86.12 | 36.69 | 62.59 | 38.81 | **56.57** |
| *Llama2-13B* | | | | | | | | | |
| $k = 16$ | 66.49 | 69.13 | 44.9 | 57.04 | 86.24 | 43.17 | 54.46 | 41.98 | **57.16** |
| $k = 32$ | 66.49 | 69.13 | 44.98 | 57.04 | 86.12 | 42.45 | 54.46 | 41.98 | 56.88 |
| $k = 64$ | 66.49 | 69.13 | 45.01 | 56.68 | 86.01 | 43.88 | 54.51 | 42.18 | 57.0 |

Table 9: Results of different mapping methods.

| Model | MRPC | COLA | MNLI | RTE | SST2 | ACL | PHRASEBANK | MUSIC | Average |
|---|---|---|---|---|---|---|---|---|---|
| *GPT-Neo-2.7B* | | | | | | | | | |
| Attention | 66.49 | 69.13 | 42.87 | 54.87 | 81.54 | 41.73 | 30.3 | 25.25 | 51.52 |
| Cosine | 66.49 | 69.13 | 40.97 | 53.79 | 85.44 | 48.92 | 31.1 | 26.73 | 52.82 |
| Pearson | 66.49 | 69.13 | 40.98 | 53.79 | 85.44 | 48.92 | 31.1 | 26.73 | 52.82 |
| Euclidean | 66.49 | 69.13 | 43.75 | 55.23 | 84.75 | 47.48 | 30.79 | 24.85 | 52.81 |
| Manhattan | 66.49 | 69.13 | 42.96 | 54.51 | 84.86 | 47.48 | 38.6 | 28.71 | **54.09** |
| *Mistral-7B* | | | | | | | | | |
| Attention | 66.49 | 69.13 | 52.07 | 52.71 | 82.22 | 51.08 | 63.21 | 42.57 | 59.93 |
| Cosine | 66.49 | 69.13 | 52.31 | 52.71 | 82.68 | 51.08 | 63.47 | 42.18 | 60.01 |
| Pearson | 66.49 | 69.13 | 52.28 | 52.71 | 82.68 | 51.08 | 63.47 | 42.18 | 60.00 |
| Euclidean | 66.49 | 69.13 | 52.01 | 52.71 | 83.26 | 51.08 | 62.37 | 42.57 | 59.95 |
| Manhattan | 66.49 | 69.13 | 52.09 | 52.71 | 85.32 | 51.08 | 63.25 | 42.87 | 60.37 |
| *Llama2-7B* | | | | | | | | | |
| Attention | 66.49 | 69.13 | 39.37 | 53.43 | 86.12 | 36.69 | 62.63 | 38.91 | 56.6 |
| Cosine | 66.49 | 69.13 | 40.36 | 53.43 | 87.16 | 41.01 | 63.43 | 39.01 | 57.5 |
| Pearson | 66.49 | 69.13 | 40.35 | 53.43 | 87.16 | 41.01 | 63.43 | 39.01 | 57.5 |
| Euclidean | 66.49 | 69.13 | 39.4 | 53.43 | 86.81 | 36.69 | 62.68 | 40.30 | 56.87 |
| Manhattan | 66.49 | 69.13 | 39.6 | 53.07 | 87.39 | 43.88 | 62.68 | 39.7 | **57.74** |
| *Llama2-13B* | | | | | | | | | |
| Attention | 66.49 | 69.13 | 45.01 | 57.04 | 86.24 | 43.88 | 54.51 | 42.18 | **57.65** |
| Cosine | 66.49 | 69.13 | 46.05 | 56.32 | 77.98 | 39.57 | 53.89 | 43.56 | 56.4 |
| Pearson | 66.49 | 69.13 | 46.05 | 56.32 | 77.98 | 39.57 | 53.89 | 43.56 | 56.4 |
| Euclidean | 66.49 | 69.13 | 48.10 | 57.40 | 79.47 | 38.85 | 53.22 | 42.97 | 56.95 |
| Manhattan | 66.49 | 69.13 | 46.79 | 57.04 | 81.88 | 35.25 | 54.55 | 42.67 | 56.73 |

