# OpenReview forum: "In-Context Learning at Representation Level via Unlabeled Texts"
_ICLR.cc/2025/Conference — Submitted to ICLR 2025_

### Official Review · Reviewer_zKpB · 2024-10-21

**Soundness:** 3
**Presentation:** 3
**Contribution:** 3
**Rating:** 6
**Confidence:** 4

**Summary:**

This paper firstly proposes two gaps between current ICL paradigm and pre-training, label appearance and weak semantic relevance. Sometimes ICL benefits from the gap but sometimes it does not. Then it conducts corresponding experiments and observe different conclusions on specific-domain and general domain datasets, based on which it proposes its method -- fusing unlabeled samples to reshape the representation of the test input for inference. This method outperforms traditional ICL on models of varying sizes.

**Strengths:**

* This paper is well-written. The motivation demonstrated by preliminary experiments is very clear.
* By viewing specific and general domain datasets separately, the observation is interesting.
* The experimental results show the method is effective.

**Weaknesses:**

* Though this method performs well on some NLU tasks, I'm curious about other diverse tasks like generation, reasoning and more difficult tasks in LLM era, since ICL can be used in many scenarios.
* I'm curious that whether getting hidden states bring extra time cost, compared with top-k example selection in traditional ICL.
* The performance on specific domain datasets is sometimes worse than the baseline. (maybe trying whether to use gold labels can be further explored)

**Questions:**

* More analysis on time cost and latency brought by this method can be investigated.
* What about the performance on other diverse tasks in LLM era, like reasoning, etc.

---

### Official Review · Reviewer_2zdv · 2024-11-02

**Soundness:** 2
**Presentation:** 2
**Contribution:** 2
**Rating:** 5
**Confidence:** 2

**Summary:**

This paper proposes a new representation-based in-context learning (ICL) paradigm, which utilizes unlabeled text in the test set for learning, without relying on annotated examples. The authors find that the presence of labels has a greater positive impact than negative impact on domain-specific datasets, but the opposite is true for general-domain datasets. Furthermore, the authors' proposed method performs inference by independently processing the representation of the example input, which is superior to the traditional ICL based on annotated examples, and allows smaller models to outperform larger models in zero-shot performance.

**Strengths:**

It proposes a new representation-based in-context learning (ICL) paradigm that utilizes the unlabeled text in the test set for learning, without relying on annotated examples.

It finds that the presence of labels has a greater positive impact than negative impact on specific domain datasets, but the opposite is true for general domain datasets.

The proposed method performs reasoning by independently processing the representations of the example inputs, which is superior to the traditional ICL based on annotated examples, and allows smaller models to outperform larger models in zero-shot performance.

**Weaknesses:**

- The paper requires some additional comparation to some in-context vector methods like [1][2], which also create a hidden state offset by in-context example.

- Required some additional ablation study to prove why use the unlabeled texts from test set rather than labelled.For example, if you use labelled data in your framework in Figure 2, how will the performance be?

- The improvement form increasing numbers of the retrieved hidden states is limited.


[1]Liu, S., Ye, H., Xing, L., & Zou, J.Y. (2023). In-context Vectors: Making In Context Learning More Effective and Controllable Through Latent Space Steering. ArXiv, abs/2311.06668.
[2]Hendel, R., Geva, M., & Globerson, A. (2023). In-Context Learning Creates Task Vectors. ArXiv, abs/2310.15916.

**Questions:**

- The link of the codes is invalid.

- Besides MRPC, the difference between labelled and unlabelled data for ICL is limited. Is there any more dataset can be used for prove the conclusion about un-labelled ICL in General-Domain.

- Could you share the prompts or give some analysis about the effect of the prompt use to get the presentation of the in-context examples?

---

> ### Author Response · Authors · 2024-11-24
>
> **Comment 1**: The paper requires some additional comparation to some in-context vector methods like [1][2], which also create a hidden state offset by in-context example.
>
> Thanks for your kind reminder. We have already considered the papers you mentioned when developing our paper. The reason for not comparing our method with those papers is that our setting differs from the papers you referenced.  We focus on utilizing unlabeled texts for in-context learning. In real-world situations, gold standard demonstrations can be difficult to come by. However, The methods mentioned in the papers you cited rely on gold labels from training data. Therefore, this difference makes a direct comparison less meaningful in our context.
>
> ---
>
> **Comment 2**: Required some additional ablation study to prove why use the unlabeled texts from test set rather than labeled. For example, if you use labeled data in your framework in Figure 2, how will the performance be?
>
> Thank you for raising an interesting question. The motivation of using unlabeled texts stems from two factors. First, obtaining gold standard demonstrations in real-world scenarios can be challenging. Therefore, there is a need to develop a method for utilizing unlabeled texts for in-context learning. Second, as shown in Table 2 on page 3, we observe an increase in ICL performance when labels are removed. Taking into account these two factors, we have decided to incorporate unlabeled texts for in-context learning. However, our method is dedicated to unlabeled texts instead of input-label pairs. We appreciate your valuable feedback of using labeled data and will investigate the potential for input-label pairs in our method in the future.
>
> ---
>
> **Comment 3**: The improvement from increasing numbers of the retrieved hidden states is limited.
>
> Thank you for careful reading of these details. The limited improvement from increasing numbers of the retrieved hidden states is more likely to be caused by the composition of the test dataset. Our method depends on the similarity between the retrieved texts with the test input to reconstruct the input test vector. However, when constructing the test dataset, we typically want the test dataset to be diverse. Therefore, it is hard to find the relevant examples to the test input. The performance can not be improved much by increasing numbers of the retrieved hidden states. We appreciate your valuable feedback and will construct more diverse datasets to better utilizing larger number of the retrieved hidden states.
>
> ---
>
> **Comment 4**: The link of the codes is invalid.
>
> Thank you for your interest in our work. We are willing to make our code open-source, but the rules of OpenReview prohibit us from including any links that reveal personal identity in papers under review. Instead, links should contain the word "anonymous" to signify that we will release our code. The complete implementation of our paper will be released in the future.
>
> ---
>
> **Comment 5**: Besides MRPC, the difference between labeled and unlabeled data for ICL is limited. Is there any more dataset can be used for prove the conclusion about un-labeled ICL in general-domain?
>
> Thank you for carefully reviewing our experiments. There are two experiments supporting the claim that labels may not be necessary for in-context learning in the general domain. In addition to the traditional ICL experiments with and without labels, the comparison between our method without labels and traditional ICL also confirms this claim. As shown in Table 6 on page 9, the largest improvement of our method without labels compared to ICL with labels is observed in the general domain. The large improvement in general domain indicates labels may not be necessary for ICL in the general domain. Combining these two experiments, the claim is reasonable.

---

> ### Author Response · Authors · 2024-11-24
>
> **Comment 6**: Could you share the prompts or give some analysis about the effect of the prompt use to get the presentation of the in-context examples?
>
> Thanks for your careful reading of these details. The prompts used in the experiments are presented in the below Table. We will modify the paper to include the prompts.
>
> | **Dataset** |                                                       **Representation Prompts**                                                       |                                             **Inference Prompts**                                            |
> |:-----------:|:--------------------------------------------------------------------------------------------------------------------------------------:|:------------------------------------------------------------------------------------------------------------:|
> |     MRPC    |                   Represent the input two sentences to better determine whether they are equivalent or not.\nsentence1: {sentence1}\tsentence2: {sentence2}                    |                         The two sentences are (not_equivalent\|equivalent).\nsentence1: {sentence1}\tsentence2: {sentence2}                         |
> |     COLA    |                        Represent the sentence to better classify whether its grammar is correct or not.\nsentence: {sentence}                       |                      The grammar of the sentence is (unacceptable\|acceptable).\nsentence: {sentence}                     |
> |     MNLI    |        Represent the premise and hypothesis to better determine their relationship (entailment, neutral, contradiction).\npremise: {premise}\thypothesis: {hypothesis}       |                     The premise and hypothesis (agree\|are unrelated\|disagree).\npremise: {premise}\thypothesis: {hypothesis}                    |
> |     RTE     |                   Represent the input two sentences to better determine whether one entails another or not.\nsentence1: {sentence1}\tsentence2: {sentence2}                  | The two sentences (have\|do not have) the relationship that if one is true, then the other is true.\nsentence1: {sentence1}\tsentence2: {sentence2} |
> |     SST2    |                       Represent the movie review to better determine whether it is positive or negative.\nsentence: {sentence}                      |                              The movie review is (negative\|positive).\nsentence: {sentence}                              |
> |     ACL     |                           Represent the paper segment to better classify the intent of the citation.\ntext: {text}                          |  The intent of the citation is (Background\|Uses\|CompareOrContrast\|Motivation\|Future\|Extends).\ntext: {text}  |
> |    PHRASE   |                          Represent the sentence from financial news to better classify its sentiment.\nsentence: {sentence}                         |                   The sentiment of the sentence is (negative\|neutral\|positive).\nsentence: {sentence}                   |
> |    MUSIC    | Represent the text to better classify the type of music it describes as jazz, country, folk, r&b, pop, rock, dance, or latin.\ntext: {text} |                The music type is (Jazz\|Country\|Folk\|R&B\|Pop\|Rock\|Dance\|Latin).\ntext: {text}               |

---

> ### Comment · Reviewer_2zdv · 2024-11-25
>
> Thank you for your response.

---

### Official Review · Reviewer_w7FD · 2024-11-03

**Soundness:** 2
**Presentation:** 2
**Contribution:** 2
**Rating:** 5
**Confidence:** 4

**Summary:**

This work proposes a new in-context learning paradigm. It is mainly motivated by the identified issue of weak semantic relevance in traditional ICL: ICL demonstrations exhibit weaker semantic dependence than the pretraining corpora. Therefore, the authors propose to process ICL examples independently for better coherence within each example. It then use the attention mechanism to integrate these independent representations of examples into the test input's representation. In addition, this method does not require labels for ICL examples. Experiments are conducted on four LLMs and eight datasets.

**Strengths:**

- This work highlights a discrepancy  between ICL and LLM pretraining - unlike coherent text used in pretraining, ICL demonstrations exhibit weaker semantic dependence
- The proposed paradigm of conducting ICL at the representation level is new.

**Weaknesses:**

- The first concern is the absence of empirical evidence to support the theoretical claims about the weak semantic relevance's impact on ICL performance. Without quantitative evaluation, it is unknown whether the impact of this discrepancy is significant, making the main motivation of this work not well-supported. Additional experiments could be conducted to compare conventional ICL demonstrations with modified ones that include semantically coherent transitions.
- The proposed ICL paradigm simplifies the interaction between examples and the test input into a single step, potentially losing vital information that could be obtained in multi-layer interactions of conventional ICL.
- The proposed method is likely to increase computational and storage demands in computing independent representations of each example and reconstructing the representation of the test sample. However, there is a lack of analysis on the efficiency of the proposed method.
- There is a lack of experimental details. For example, the prompts used in Section 2 and the main experiments are not provided, and the methodology for selecting hyperparameters in Equation 11 is not introduced, and the references of the baselines used in the main experiment are missing.
- Some discrepancies in ICL performance of baselines are observed between this paper and the literature. For example, the 16-shot ICL performance using random examples for Llama2-7B on SST-2, RTE, and CoLA is reported as 93.16, 77.02, and 70.20, respectively in [1], which is 20-30 points higher than that reported in this study.

---
[1] Van, Minh-Hao, and Xintao Wu. "In-Context Learning Demonstration Selection via Influence Analysis." arXiv preprint arXiv:2402.11750 (2024).

**Questions:**

- Why does the proposed method achieve the same accuracy across four different LLMs on MRPC and CoLA datasets?
- Why there is a decrease in performance when upgrading from Llama2-7B to Llama2-13B on SST-2 and Phrase datasets using the proposed method?

---

> ### Author Response · Authors · 2024-11-24
>
> **Comment 1**: The first concern is the absence of empirical evidence to support the theoretical claims about the weak semantic relevance's impact on ICL performance.
>
> The empirical evidence to support the claims about the weak semantic relevance's impact on ICL performance can be found in Table 5 on page 8. The traditional ICL takes in the concatenation of multiple demonstrations which are not necessarily relevant. We compare traditional ICL to our method that processes demonstration inputs and test input independently. As shown in Table 5, the traditional concatenation performance for GPT-Neo-2.7B and Llama2-7B is at least $7.94$ and $10.57$ worse than representing the inputs independently. The notable performance drop for two models indicates the negative impact of weak semantic relevance caused by the traditional concatenation, aligning with the theoretical claims about the weak semantic relevance's impact on ICL performance.
>
> ----
>
> **Comment 2**: The proposed ICL paradigm simplifies the interaction between examples and the test input into a single step, potentially losing vital information that could be obtained in multi-layer interactions of conventional ICL.
>
> Your concern about our method potentially losing vital information is reasonable. We took this into consideration when developing our method. The most common practice is to use the performance metric for assessing whether a method effectively retains vital information or not. The results in Table 6 demonstrate that our method outperforms traditional in-context learning for GPT-Neo-2.7B, Mistral-7B, Llama2-7B, and Llama2-13B in three well-known settings. Despite simplifying the interaction process between examples and test input into a single step, our method still brings significant improvements. This suggests that our method may uncover crucial insights that traditional ICL methods may miss. This finding underscores the importance of further exploration within the ICL community. We appreciate your valuable feedback and will investigate the potential for multi-layer interaction in our method in the future.
>
> ---
>
> **Comment 3**: The proposed method is likely to increase computational and storage demands in computing independent representations of each example and reconstructing the representation of the test sample.
>
> 1. The computational demands for computing independent representations for each example and reconstructing the representation of the test sample are much lower than the traditional ICL. With our approach, the model only needs to process one example at a time instead of the $k$ concatenated examples, resulting in lower GPU memory usage. The computing latency caused by our method is also within an acceptable range.
> 2. While it is true that we require memory for storing representation vectors, advances in text retrieval have succeeded in reducing the memory required for vectors, such as through optimized vector databases. Therefore, when using the techniques in these vector databases, the memory will not be large.
> 3. When applying our method in real-world scenarios, such as determining whether the intent of a user is positive or negative, we find that our approach is faster than traditional ICL because we pre-compute the representation vectors ahead of time.

---

> ### Author Response · Authors · 2024-11-24
>
> **Comment 4**: For example, the prompts used in Section 2 and the main experiments are not provided, and the methodology for selecting hyperparameters in Equation 11 is not introduced, and the references of the baselines used in the main experiment are missing.
>
> Thanks for your careful reading of these details.
>
> 1. The prompts used in the experiments are presented in the below Table. We will modify the paper to include the prompts.
> 2. We will add the discussion of selecting hyperparameters in Equation 11 to our paper. The selection relies on the intuition and some simple experiments. We discover that it is important to assign larger weights to the representation of the original test input, rather than giving the same weights to them.
> 3. The references of the baselines used in the main experiment is in **Section 2.1 Analysis Settings** on page 2. From your comment, we realize it may be not friendly for readers to find the baselines. We will add the references of the baselines in every experiment section. Specifically, we compare our method with zero-shot inference and traditional ICL [1] with gold input-label pairs from the training set.
>
> [1] Tom Brown, Benjamin Mann, Nick Ryder, Melanie Subbiah, Jared D Kaplan, Prafulla Dhariwal, Arvind Neelakantan, Pranav Shyam, Girish Sastry, Amanda Askell, et al. Language models are few-shot learners. Advances in neural information processing systems, 33:1877–1901, 2020.
>
> | **Dataset** |                                                       **Representation Prompts**                                                       |                                             **Inference Prompts**                                            |
> |:-----------:|:--------------------------------------------------------------------------------------------------------------------------------------:|:------------------------------------------------------------------------------------------------------------:|
> |     MRPC    |                   Represent the input two sentences to better determine whether they are equivalent or not.\nsentence1: {sentence1}\tsentence2: {sentence2}                    |                         The two sentences are (not_equivalent\|equivalent).\nsentence1: {sentence1}\tsentence2: {sentence2}                         |
> |     COLA    |                        Represent the sentence to better classify whether its grammar is correct or not.\nsentence: {sentence}                       |                      The grammar of the sentence is (unacceptable\|acceptable).\nsentence: {sentence}                     |
> |     MNLI    |        Represent the premise and hypothesis to better determine their relationship (entailment, neutral, contradiction).\npremise: {premise}\thypothesis: {hypothesis}       |                     The premise and hypothesis (agree\|are unrelated\|disagree).\npremise: {premise}\thypothesis: {hypothesis}                    |
> |     RTE     |                   Represent the input two sentences to better determine whether one entails another or not.\nsentence1: {sentence1}\tsentence2: {sentence2}                  | The two sentences (have\|do not have) the relationship that if one is true, then the other is true.\nsentence1: {sentence1}\tsentence2: {sentence2} |
> |     SST2    |                       Represent the movie review to better determine whether it is positive or negative.\nsentence: {sentence}                      |                              The movie review is (negative\|positive).\nsentence: {sentence}                              |
> |     ACL     |                           Represent the paper segment to better classify the intent of the citation.\ntext: {text}                          |  The intent of the citation is (Background\|Uses\|CompareOrContrast\|Motivation\|Future\|Extends).\ntext: {text}  |
> |    PHRASE   |                          Represent the sentence from financial news to better classify its sentiment.\nsentence: {sentence}                         |                   The sentiment of the sentence is (negative\|neutral\|positive).\nsentence: {sentence}                   |
> |    MUSIC    | Represent the text to better classify the type of music it describes as jazz, country, folk, r&b, pop, rock, dance, or latin.\ntext: {text} |                The music type is (Jazz\|Country\|Folk\|R&B\|Pop\|Rock\|Dance\|Latin).\ntext: {text}               |

---

> ### Author Response · Authors · 2024-11-24
>
> **Comment 5**: Some discrepancies in ICL performance of baselines are observed between this paper and the literature.
>
> Thanks for your concern about the experimental results. The ICL performance in our paper is produced by the framework proposed in [OpenICL: An Open-Source Framework for In-context Learning](https://aclanthology.org/2023.acl-demo.47/) presented at ACL 2023. This framework provides standard implementation of ICL and has been widely adopted in other ICL papers [1][2][3], demonstrating its reliability. Also, the framework is widely recognized in the ICL community (542 stars on GitHub). Therefore, the results of our paper are reliable.
>
> Regarding the differences with the referenced paper, we carefully check the implementation of the referenced paper. The following reasons may account for the performance difference.
>
> 1. In-context learning is quite unstable to the prompts. Our prompts can be quite different with those in that paper. For instance, for rte dataset, they used "{premise}\nquestion: {hypothesis}. true or false?\n" while we used "The two sentences  (have | do not have) the relationship that if one is true, then the other is true.\nsentence1: {sentence1}\tsentence2: {sentence2}". Different prompts can result in quite different results.
> 2. They loaded the model with half precision to save the memory, while we used full precision.
> 3. They **incorrectly** implemented the padding_side in the tokenizer. When calculating the perplexity loss for the sequence, the padding_side should be right, not left. With padding_side set to left, language models are conditioned on the <pad> tokens when processing the entire sequence, while <pad> tokens are not seen by LLMs during pretraining. The padding_side should be left when language models are required to generate new tokens.
>
> To summarize, the above three factors can greatly impact the final performance. Therefore, the performance discrepancies should be reasonable.
>
> [1] **ACL 2024** | Forward-Backward Reasoning in Large Language Models for Mathematical Verification by Weisen Jiang, Han Shi, Longhui Yu, Zhengying Liu, Yu Zhang, Zhenguo Li, James Kwok
>
> [2] **NAACL 2024** | Multilingual Machine Translation with Large Language Models: Empirical Results and Analysis by Wenhao Zhu, Hongyi Liu, Qingxiu Dong, Jingjing Xu, Shujian Huang, Lingpeng Kong, Jiajun Chen, Lei Li.
>
> [3] **ICLR 2024**  | OS-Copilot: Towards Generalist Computer Agents with Self-Improvement by Zhiyong Wu, Chengcheng Han, Zichen Ding, Zhenmin Weng, Zhoumianze Liu, Shunyu Yao, Tao Yu, Lingpeng Kong
>
> ----
>
> **Q1**: Why does the proposed method achieve the same accuracy across four different LLMs on MRPC and CoLA datasets?
>
> The MRPC and CoLA datasets are both from the general domain. It is probable that both datasets are pretrained by four models. Therefore, the pretrained knowledge of four models on the MRPC and CoLA datasets is likely to be similar.
>
> ---
>
> **Q2**: Why there is a decrease in performance when upgrading from Llama2-7B to Llama2-13B on SST-2 and Phrase datasets using the proposed method?
>
> The decrease in performance is due to the weaker abilities of Llama2-13B compared to Llama2-7B for the performance decrease is also observed in zero-shot when upgrading to Llama2-13B.

---

> > ### Comment · Reviewer_w7FD · 2024-11-26
> >
> > I appreciate the authors' responses during the rebuttal.
> >
> > **Responses to comment 1,2**:
> >
> > I understand that the authors aim to respond to these two concerns by highlighting the performance improvements of their proposed method. I agree this does address my concerns to some extent. However, a limitation of this performance-driven proof is that it demonstrates effectiveness solely through task performance without providing in-depth insights into why the method works.
> >
> > **Responses to comment 3**:
> >
> > It would be better to quantify the claims with empirical evidence for statements such as "The computing latency caused by our method is also within an acceptable range," "When using the techniques in these vector databases, the memory will not be large"
> >
> > **Responses to comment 4**:
> >
> > Thanks for providing the prompts. Please include them and the hyperparameter selection method in the paper.
> >
> > **Responses to comment 5**:
> >
> >  appreciate the authors for detailing how they produced their experimental results. However, the discrepancy in baseline results (randomly selecting ICL examples) is too large—20-30 percentage points lower compared to the literature [1-3]. While I would like to trust the experimental results presented in this paper, it remains unclear if the method would be effective when the baseline adopts prompts from these references and achieves significantly better accuracy.
> >
> > I will increase my score to 5. I appreciate the novelty of this work. However, I maintain my concerns regarding the empirical validation of the method and the lack of in-depth empirical evidence to support why the method works. Therefore, I assign a score below the acceptance threshold.
> >
> > ---
> > [1] Van, Minh-Hao, and Xintao Wu. "In-Context Learning Demonstration Selection via Influence Analysis." arXiv preprint arXiv:2402.11750 (2024)
> >
> > [2] Voronov, Anton, Lena Wolf, and Max Ryabinin. "Mind your format: Towards consistent evaluation of in-context learning improvements." arXiv preprint arXiv:2401.06766 (2024).
> >
> > [3] Peng, Keqin, et al. "Revisiting demonstration selection strategies in in-context learning." arXiv preprint arXiv:2401.12087 (2024).

---

### Official Review · Reviewer_ismB · 2024-11-06

**Soundness:** 2
**Presentation:** 2
**Contribution:** 2
**Rating:** 5
**Confidence:** 3

**Summary:**

This paper introduces a novel in-context learning (ICL) method at the representation level, leveraging (unlabeled) sentences from the test set.

The proposed method is motivated by two existing gaps between pre-training and ICL:
1. **Label appearance**: During pre-training, the text lacks task-specific signals, while in ICL, the input-label mapping introduces explicit task information.
2. **Weak semantic relevance**: Text used in pre-training tends to be more coherent, whereas ICL demonstrations are often relatively unrelated to each other.

To address these issues, the proposed approach builds on **unlabeled ICL**.
Specifically, representations of different test sentences are first pre-computed using an LLM.
The new test input is then encoded with the same model, and this representation is used as a query in the attention mechanism, with $k$ relevant representations from the previous stage functioning as keys and values.
Finally, the attention mechanism’s output vectors for the $k$ different samples are averaged and combined with the original test input feature vector.
This final vector serves as input for the target language model’s lm_head, where the likelihood of each option is computed, and the option with the highest likelihood is selected as the final answer.

In the experiments, the proposed method is compared with zero-shot and few-shot ICL. Results indicate that it generally outperforms zero-shot and is comparable to other demonstration selection-based few-shot baselines.

**Strengths:**

- The proposed method does not rely on label information from demonstrations, making it applicable in cases where only relevant context is available without gold-standard labels.
- Interestingly, the proposed method can be interpreted as representing the vector of the target input within the space spanned by vectors of other samples.
- An analysis was conducted to examine the inner workings of ICL from the authors’ own perspectives, providing insights into the paradigm.

**Weaknesses:**

- While assumptions like label appearance and weak semantic relevance are intriguing, I am somewhat doubtful, especially regarding weak semantic relevance. In pre-training, language models are exposed to a substantial variety of cases, some of which might closely resemble in-context learning scenarios. For instance, if multi-choice QA datasets were included in pre-training, the answer choices could introduce sequences that appear relatively unrelated. Thus, it may not be easy to guarantee that LMs are unfamiliar with inputs seen in in-context learning scenarios.
- I’m also unsure whether comparison with only zero-shot ICL is entirely fair. Although the proposed method does not utilize label information from the test set, it does use textual information from it. Could we add more reliable baselines, such as performing few-shot ICL with random labels (distinct from a random few-shot retrieval-based baseline) or using self-generated labels?
- Equation 11 seems somewhat arbitrary, with no explanation provided for why those specific hyperparameters (e.g., 0.4 and 0.6) are applied in the weighted sum. Could you elaborate on this process?
- It would be helpful if the paper included an analysis of the proposed method’s efficiency. While accuracy is critical for evaluating performance, the method’s efficiency is also a key factor in understanding its practical implications.

**Questions:**

- I am curious why sentences from the test set are used to construct representations for the method’s computation. What would happen if we relied on sentences from the training set instead? Although the proposed method does not rely on label information from the test set, avoiding the use of any hints or information extractable from the test set, if possible, would help ensure a fair evaluation.

---

> ### Author Response · Authors · 2024-11-24
>
> **Comment 1**: While assumptions like label appearance and weak semantic relevance are intriguing, I am somewhat doubtful, especially regarding weak semantic relevance. In pre-training, language models are exposed to a substantial variety of cases, some of which might closely resemble in-context learning scenarios. For instance, if multi-choice QA datasets were included in pre-training, the answer choices could introduce sequences that appear relatively unrelated. Thus, it may not be easy to guarantee that LMs are unfamiliar with inputs seen in in-context learning scenarios.
>
> We appreciate your careful reading of this assumption. We also consider this assumption when developing this method. We provide the following evidence to support that LMs may be unfamiliar with inputs in in-context learning scenarios and need better methods to handle the weak semantic relevance between demonstrations.
>
> 1. While it is true that in-context learning style datasets like QA datasets may be included in pre-training, the proportion of ICL datasets is relatively small compared to the regular coherent texts. Therefore, LLMs are more likely to remember the regular coherent texts instead of ICL style datasets during pretraining.
> 2. If LLMs are already acquainted with the inputs in ICL scenarios, changing the order of the demonstrations should not have much impact on performance. In reality, however, performance is actually quite sensitive to the order.
> 3. The empirical evidence to support the claims about the weak semantic relevance's impact on ICL performance can be found in Table 5 on page 8. The traditional ICL takes in the concatenation of multiple demonstrations which are not necessarily relevant. We compare traditional ICL to our method that processes demonstration inputs and test input independently. As shown in Table 5, the traditional concatenation performance for GPT-Neo-2.7B and Llama2-7B is at least $7.94$ and $10.57$ worse than representing the inputs independently. The notable performance drop for two models indicates the negative impact of weak semantic relevance caused by the traditional concatenation.
>
> ---
>
> **Comment 2**: Although the proposed method does not utilize label information from the test set, it does use textual information from it. Could we add more reliable baselines, such as performing few-shot ICL with random labels (distinct from a random few-shot retrieval-based baseline) or using self-generated labels?
>
> Thanks for your kind reminder. In Table 6 on page 9, we have already compared our method with the few-shot ICL with gold labels from the training set. The results in Table 6 demonstrate that our method outperforms traditional in-context learning for GPT-Neo-2.7B, Mistral-7B, Llama2-7B, and Llama2-13B in three well-known settings. From your comment, we realize that it may be not easy for readers to find the comparison between our method and few-shot ICL. We will add a brief outline at the beginning of the experimental section to make it easy for readers to check different experiments.
>
> ---
>
> **Comment 3**: Equation 11 seems somewhat arbitrary, with no explanation provided for why those specific hyperparameters (e.g., 0.4 and 0.6) are applied in the weighted sum. Could you elaborate on this process?
>
> Thanks for your careful reading of the details. We will add the discussion of selecting hyperparameters in Equation 11 to our paper. The selection relies on the intuition and some simple experiments. We discover that it is important to assign larger weights to the representation of the original test input, rather than giving the same weights to them.

---

> ### Author Response · Authors · 2024-11-24
>
> **Comment 4**: It would be helpful if the paper included an analysis of the proposed method’s efficiency. While accuracy is critical for evaluating performance, the method’s efficiency is also a key factor in understanding its practical implications.
>
> Your reminder is helpful for readers to better understand our method. We will include a more detailed efficiency analysis in the paper.
> 1. **Time Efficiency**: When conducting per-dataset testing, our method is slightly slower than traditional ICL due to the retrieving of unlabeled examples. Therefore, we choose BM25 in this paper for its efficiency. However, when applying our method in real-world scenarios, such as determining whether the intent of a user is positive or negative, we find that our approach is faster than traditional ICL because we pre-compute the representation vectors ahead of time.
> 2. **Computational Efficiency**: The computational demands for computing independent representations for each example and reconstructing the representation of the test sample are much lower than the traditional ICL. With our approach, the model only needs to process one example at a time instead of the $k$ concatenated examples, resulting in lower GPU memory usage. The computing latency caused by our method is also within an acceptable range.
> 3. **Memory Efficiency**: While it is true that we require memory for storing representation vectors, advances in text retrieval have succeeded in reducing the memory required for vectors, such as through optimized vector databases. Therefore, when using the techniques in these vector databases, the memory will not be large.
>
> ----
>
> **Comment 5**: I am curious why sentences from the test set are used to construct representations for the method’s computation.
>
> Thank you for raising an interesting question. The motivation of using unlabeled texts from the test dataset stems from two factors. First, obtaining gold standard demonstrations in real-world scenarios can be challenging. Therefore, there is a need to develop a method for utilizing unlabeled texts for in-context learning. Second, as shown in Table 2 on page 3, we observe an increase in ICL performance when labels are removed. Taking into account these two factors, we have decided to incorporate unlabeled texts for in-context learning.

---

> ### Author Response · Authors · 2024-12-03
>
> Dear Reviewer,
>
> Thank you for the time and effort you have dedicated to reviewing our paper. We sincerely appreciate your insightful comments and suggestions. Your comments have been highly valuable to us, and we have put effort into addressing your concern. We would love to receive feedback from you. If your concern is addressed, we humbly invite the reviewer to consider increasing the score. Your support is deeply appreciated!

---

### Meta-Review · Area_Chair_bwF7 · 2024-12-20

**Metareview:**

This paper proposes a new zero-shot in-context learning (ICL) method, namely "ICL at the representation level", which uses unlabeled input text from the test dataset. The approach is motivated by two key gaps between pre-training and ICL: the absence of labels and the lack of coherence in the input. The method encodes each unlabeled text and test input independently, computes dependencies at the representation level, and generates predictions. Results demonstrate up to 17% improvement over the baseline zero-shot method, and even outperform standard ICL methods.

Strengths
- Introduction of a new method for an important problem (zero-shot ICL) with good empirical results (ismB, w7FD, 2zdv, zKpB)
- The proposed method can be interpretable (ismB)
- Insightful analysis (ismB, 2zdv, zKpB)
- The paper is well written (zKpB)

Weaknesses
- Assumptions in the motivations are not very convincing, e.g., for instance, the structure of the ICL may actually frequently appear in pre-training (ismB, w7FD).
- The fact that the proposed method uses unlabeled test data makes empirical comparison unfair (ismB).
    - Responses from authors point to Table 6 that compare with standard ICL. However, as standard ICL also does not use unlabeled test data, it doesn’t completely solve the problem.
- Lack of discussion on efficiency (ismB).
- Lack of experimental details (w7FD)
- Baseline numbers significantly lower than results from prior work (20-30%) (w7FD)
    - Author responses point out differences to prior work and that prior work has some issues. However, this only explains why prior work's results might be worse, whereas, here, prior work reports higher numbers. Also, the concern that the improvement might be due to a weak baseline still exists.
- Lack of ablation, such as using the proposed method (ICL at the representation level) but with labels (2zdv, zKpB), or other zero-shot ICL methods like ICL with random labels and ICL with self-generated labels (ismB).

**Additional Comments On Reviewer Discussion:**

In AC's opinion, authors' responses during the rebuttal period did not sufficiently address reviewers' concerns.

---

### Decision · Program_Chairs · 2025-01-22

Reject